# New Insights into the Mechanism of Spatiotemporal Scent Accumulation in Orchid Flowers

**DOI:** 10.3390/plants12020304

**Published:** 2023-01-09

**Authors:** Bao-Qiang Zheng, Xiao-Qing Li, Yan Wang

**Affiliations:** Key Laboratory of Tree Breeding and Cultivation of State Forestry Administration, Research Institute of Forestry, Chinese Academy of Forestry, Beijing 100091, China

**Keywords:** floral scent, mechanism, orchid, monoterpene, flower

## Abstract

Orchid flowers have a unique structure that consists of three sepals and three petals, with one of the petals forming the labellum (lip) that can be differentiated into the hypochile and epichile. In orchids, the emission of floral scent is specific and spatially complex. Little is understood about the molecular and biochemical mechanisms of the differing scent emissions between the parts of orchid flowers. Here, we investigated this in the *Cattleya* hybrid KOVA, and our study showed that monoterpenes, including linalool and geraniol, are the main components responsible for the KOVA floral scent. The KOVA flower was scentless to the human nose before it reached full bloom, potentially because the *1-deoxy-d-xylulose 5-phosphate synthase*s (*RcDXS*s) and *4-hydroxy-3-methylbut-2-enyl diphosphate synthase*s (*RcHDS*s) that biosynthesize monoterpenes were highly expressed in flowers only when it reached full flowering. Additionally, the spatial expression profile of the *monoterpene synthases* (*RcMTPS*s), which were highly expressed in the basal region of the lip (hypochile), contributed to the highest monoterpene emissions from this part of the flower. This might have caused the hypochile to be more fragrant than the other parts of the flower. These findings enrich our understanding of the difference in scents between different flower parts in plants and provide information to breed novel orchid cultivars with special floral scents.

## 1. Introduction

Flower scents can help plants to attract pollinators [1] and are an important commodity for ornamental flowers [2,3,4]. Thus, the biochemical mechanism of flower scent formation is an attractive topic to scientists and breeders and has been widely described for many plants [5,6,7,8]. Orchids are one of the most famous plants worldwide due to their beautiful, colorful, and fragrant flowers, with three sepals and three petals, of which one is differentiated into the lip (labellum) that is unique [9]. There is limited understanding of the composition and intensity of scent volatiles among different flower parts, although the underlying biochemistry of volatile synthesis has been investigated in some *Phalaenopsis* and *Dendrobium* orchids [10,11,12,13,14].

Terpenoids, one of the main components of floral scents, are classified into several types and consist mainly of sesquiterpenes, monoterpenes, diterpenes, and volatile carotenoid derivatives. These groups are classified according to the number of five-carbon (C_5_) units and their biosynthesis via either the cytosolic mevalonic acid (MVA) or the plastidial methylerythritol phosphate (MEP) pathways [15]. Monoterpenes have been found to be especially important floral volatile organic compounds (VOCs) in plants [3,16,17,18]. They also act as the main compounds of floral scents in orchid species. Linalool and geraniol and their derivatives, *beta*-Ocimene and myrcene, determine the floral scent of *Phalaenopsis bellina*, whereas these monoterpenes are not found in the scentless species *P*. *equestris* and *P*. *aphrodite* [10,13]. The floral scent of the fragrant *Phalaenopsis* orchid *P*. I-Hsin Venus KHM221 is dominated by three monoterpenes, including linalool, geraniol, and eucalyptol [12]. Additionally, linalool and geraniol have also been indicated to be the main VOCs that determine the floral scent of *Dendrobium officinale* [19]. Based on these previous studies, linalool and geraniol are common VOCs emitted by flowers in *Phalaenopsis* and *Dendrobium* species, and there are differences in the types of other monoterpenes in different orchid species.

Monoterpenes are products of the MEP pathway, which has been clearly described and mainly consists of ten enzymes: 1-deoxy-d-xylulose 5-phosphate synthase (DXS), 1-deoxy-d-xylulose-5-phosphate reductoisomerase (DXR), 2-C-methyl-d-erythritol 4-phosphate cytidylyltransferase (MCT), 4-(cytidine 5′-diphospho)-2-C-methyl-d-erythritol kinase (CMK), 2-C-methyl-d-erythritol 2,4-cyclodiphosphate synthase (MDS), 4-hydroxy-3-methylbut-2-enyl diphosphate synthase (HDS), 1-hydroxy-2-methyl-2-*trans*-butenyl 4-diphosphate reductase (HDR), isopentenyl diphosphate isomerase (IDI), geranyl diphosphate synthase (GDPS), and terpene synthase (TPS) [15,20]. In the MEP pathway, monoterpene synthases (MTPSs) belonging to the TPS family are involved in the final step of catalyzing geranyl pyrophosphate (GPP) to produce monoterpenes [21]. MTPSs are regarded as among the important enzymes determining the types and contents of monoterpenes in flowers [15,22]. In *Freesia* × *hybrida*, the spatiotemporal expression of *FhTPS1* plays an important role in floral scent and is responsible for linalool biosynthesis [17]. *Linalool synthase*, an *MTPS* from *Clarkia breweri*, was found to be overexpressed in carnations (*Dianthus caryophyllus* ‘Eilat’), which resulted in the emission of linalool and its derivatives from the leaves and flowers of transgenic carnation lines [23]. Additionally, the transient overexpression of *PbTPS10* and *PbTPS5* identified from *Phalaenopsis bellina* in *P*. I-Hsin flowers increased the emission of linalool [24].

*Rhyncholaeliocattleya* Beauty Girl ‘KOVA’ (KOVA), with strongly scented flowers, is an ornamental orchid cultivar cultivated in South America and is a cross of *Rhyncholaelia* species and *Cattleya* species. Based on perception by the human nose, it has a scentless gynostemium and the scent of its yellow hypochile (the basal region of the lip) is stronger than that emitted by the purple–red epichile (the apical region of the lip), sepals, and other petals (Figure 1). In this study, we aim to describe the spatiotemporal patterns of (1) scent emission in KOVA flowers based on headspace scent collection and GC-MS analysis and (2) the biosynthetic pathways of the scent volatiles using transcriptome analyses. We found that monoterpenes, including linalool and geraniol, were the main contributors to floral scent. The unique expression profiles of *RhMTPS*s might have caused the high accumulation of several monoterpenes in the hypochile, including nerol, D-limonene, citronellol, and (-)-*trans*-Isopiperitenol, which might have been the main reason why the hypochile was more fragrant than the other flower parts.

## 2. Results

### 2.1. The VOCs Emitted by KOVA Flowers

Two benzenoids, including benzaldehyde and benzyl alcohol, a sesquiterpene (*beta*-Farnesene), and several monoterpenes, including linalool, *beta*-Ocimene, *cis*-allo-Ocimene, *cis*-Citral, citral, and geraniol, were detected in the KOVA floral scent during D7 and D8 (Figure 2). The results showed that two benzenoids, *beta*-Farnesene, and linalool were detected from KOVA flowers at both stages D7 and D8, and the relative contents of benzenoids and *beta*-Farnesene emitted by KOVA flowers at D7 were obviously higher than those at D8, while the relative content of linalool showed the opposite trend. Additionally, five monoterpenes, including geraniol, *beta*-Ocimene, *cis*-allo-Ocimene, *cis*-Citral, and citral, were only emitted by flowers at D8 (Figure 2). These results suggest that monoterpenes are the main VOCs constituting the floral scent emitted by KOVA flowers at the full flowering stage (i.e., D8).

### 2.2. Spatiotemporal Quantification of VOCs in KOVA Flowers

Seven monoterpenes, nerol, D-limonene, linalool, geraniol, (-)-*trans*-Isopiperitenol, carvone, and citronellol, were identified from the flower tissues, and the total relative contents of monoterpenes gradually increased in each part during flowering (Figure 3A). Among these flower parts, the total relative content of monoterpenes in the hypochile at D8 was the highest, and linalool and geraniol represented the highest proportions of the total monoterpene contents, accounting for ~38% and 40%, respectively, which were lower than those in the sepal and petal (Figure 3B). However, the relative contents of nerol, D-limonene, (-)-*trans*-Isopiperitenol, and citronellol, in the total monoterpene content accounted for proportions of ~5%, 8%, 2%, and 7%, respectively, in the hypochile at D8, which were higher than those in the other flower parts (Figure 3B). Some monoterpenes, including nerol, D-limonene, (-)-*trans*-Isopiperitenol, and citronellol, were only detected in KOVA flower tissues, while *beta*-Ocimene, *cis*-allo-Ocimene, *cis*-Citral, and citral were also not detected (Figure 2 and Figure 3). Additionally, the relative contents of linalool and geraniol in four flower parts increased from stages D7 to D8 (Figure 3), which was in accordance with the variation tendency of those emitted by KOVA flowers between the two stages (Figure 2).

Most of these monoterpenes were present in each part of the KOVA flower at D8, except for carvone, which was only detected in the hypochile at D7 (Figure 4). Additionally, the relative content of each monoterpene was highest in the hypochile. The relative contents of nerol, D-limonene, and (-)-*trans*-Isopiperitenol at stage D8 were notably higher in the hypochile than those in the other parts of the flower (Figure 4).

### 2.3. Expression Profiles of the Genes Involved in the MEP Pathway in the KOVA Flower Parts during Floral Development

Given that monoterpenes are the main contributors to the floral scent of KOVA, we used transcriptome data produced by a previous study [25] to analyze the spatiotemporal expression profiles of the MEP pathway structural genes in the four flower parts during development. Most MEP pathway structural genes were not activated in the KOVA flowers at the bub stages, including D1 (developmental stage 1; i.e., bub length <2 cm) and D4 (developmental stage 4; i.e., bub length 4–5 cm), but they were highly expressed at D7 and D8 (Figure 5). Among these MEP pathway structural genes, the expression of *DXS*s encoding the rate-limiting enzyme and *HDS*s showed strong temporal specificity in that they only had high expression levels in the flowers at D8 (Figure 5), which corresponded to the increased content of monoterpenes in flowers from D7 to D8 (Figure 2 and Figure 3). In addition, the expression profiles of the *MTP*s encoding the enzyme involved in the last step of monoterpene biosynthesis were clearly spatiotemporal, and they maintained their high levels in the hypochile from D4 to D7 compared with their levels in other parts (Figure 5). These data were consistent with the finding that the contents of monoterpenes in the hypochile were higher than those in the other flower parts (Figure 2 and Figure 4).

### 2.4. Phylogenetic Analysis of MTPSs Expressed in KOVA Flowers

A phylogenetic tree based on the neighbor-joining (NJ) method was used to analyze the relationships between *Rhyncholaeliocattleya* MTPSs (RcMTPSs) and MTPSs from other plants (Figure 6). All RcMTPSs had the closest relationship with FhLin (*Freesia hybrid* linalool synthase), and their clade also included TPS-CIN (*Arabidopsis thaliana* terpene synthase-like sequence-1,8-cineole), SlTPS38 (*Solanum lycopersicum* terpene synthase 38), QH5 ((3R)-linalool synthase), CS (*trans*-beta-Ocimene synthase), and VvaTerp (*Vitis vinifera* (-)-alpha-terpineol synthase). These data suggest that RcMTPSs have similar functions to the abovementioned synthases from other plants and support the identification of several kinds of monoterpenes in the tissues of KOVA flowers (Figure 2, Figure 3 and Figure 4).

## 3. Discussion

Monoterpenes, important VOCs contributing to floral scents, have been found in many plants [20,22,26,27,28,29]. Additionally, monoterpenes, including linalool, geraniol, and several others, have been suggested to be the main VOCs emitted by *Phalaenopsis* spp. [10,12,13], *D*. *officinale* [19], and *Cymbidium ensifolium* [30]. Among the VOCs emitted by KOVA flowers, only monoterpenes, including linalool, *beta*-Ocimene, *cis*-allo-Ocimene, *cis*-Citral, citral, and geraniol, increased quantitatively from the scentless stage D7 to the scented stage D8, while the levels of benzaldehyde, benzyl alcohol, and *beta*-Farnesene showed a tendency to decline (Figure 2), which was not consistent with the change in the intensity of the floral scent based on perception by the human nose between the two stages. Although relatively high contents of benzaldehyde, benzyl alcohol, and *beta*-Farnesene were detected in the KOVA flower at stage D7, they could not reach the threshold value for perception by the human nose, which might cause the KOVA flower at D7 to be scentless to humans. Therefore, monoterpenes are the main VOCs that contribute to the scent of KOVA flowers. Additionally, at the D8 stage of bloom, the contents of several monoterpenes, including linalool, geraniol, nerol, D-limonene, and (-)-*trans*-Isopiperitenol, were higher in the tissues of the hypochile region of the lip than in the other flower parts, and the linalool and geraniol accounted for the highest proportions of the total monoterpene contents in all flower parts, respectively, at stage D8 (Figure 4). There were some differences in the compositions of emitted and endogenous scents (Figure 2 and Figure 3), which is a common phenomenon and can be explained by the different detection methods used [31,32,33]. These results suggested that linalool and geraniol were the main contributors to KOVA floral scents, and the highest production of monoterpenes in the hypochile caused this flower part to be more fragrant than the other parts at full bloom (Figure 2, Figure 3 and Figure 4). Floral scents usually help plants to attract and guide pollinators [34,35], and the lip in orchid flowers acts as an attractant and landing stage for pollinators via its visual and tactile cues and fragrance [36,37,38]. The hypochile is the basal part of the lip and near the gynostemium (Figure 1), and this suggests that the hypochile, which is the most fragrant flower part due to the increased production of monoterpenes, may facilitate the attraction of effective pollinators in the wild.

MEP pathway structural genes encoding enzymes play an important role in terpene production in flowers [15]. DXS, as the rate-limiting enzyme in the MEP pathway, can control the total content of terpenes in plants [39]. The expression levels of *Rhyncholaeliocattleya DXS*s (*RcDXS*s) were upregulated in four parts of the KOVA flowers at D8 (Figure 5). It is at this stage that most monoterpenes are biosynthesized and the total monoterpene content is the highest (Figure 2 and Figure 3). Additionally, the expressional level of another important MEP pathway gene, *Rhyncholaeliocattleya HDS* (*RcHDS*), was obviously higher in all KOVA flower parts at stage D8 than at stage D7 (Figure 5). Thus, this temporal specificity of the *RcDXS*s and *RcHDS*s expression profiles may be the main reason for KOVA flowers producing fragrance at D8. On the other hand, MTPSs, as the enzymes involved in the final step of the MEP pathway, determine the production and types of monoterpenes in flowers. ama0c15, a myrcene synthase, has been found to be expressed at a high level in flowers and contributes to myrcene emissions from *Antirrhinum majus* flowers [22]. AmNES/LIS-2 (*A. majus* nerolidol/linalool synthases-2) is located in plastids and is responsible for linalool formation in *A. majus* flowers [40]. Transiently overexpressing *PbTPS5* or *PbTPS10* from *P*. *bellina* in *Phalaenopsis* cultivar *P*. I-Hsin Venus flowers increased linalool production by 7.7- or 29.0-fold compared with the control, respectively [24]. However, there is limited understanding of the spatiotemporal expression pattern of *MTPS* in different perianth parts. In KOVA flowers, the total monoterpene content and the expression levels of *RcMTPS*s were the highest in the hypochile (Figure 3, Figure 4 and Figure 5). These results indicate that the expression of *RcMTPS*s might be the main reason that the monoterpene emissions from the hypochile were higher than those from the other parts.

In summary, the high expression of *RcDXS*s and *RcHDS*s in KOVA flowers at D8 may be responsible for the high amounts of monoterpenes biosynthesized during the full flowering at stage D8. Additionally, the higher emissions of monoterpenes from the hypochile compared with the other flower parts may result in the hypochile possibly being more fragrant to pollinators. These findings provide new insights into the biochemical and molecular mechanisms of the spatial differences in floral scents between different flower parts, which offers some clues for breeding novel orchid cultivars with special floral scents. Finally, the specific functions of *RcDXS*s, *RcHDS*s, and *RcMTPS*s in monoterpenes biosynthesis in orchid flowers need to be further investigated in the future.

## 4. Materials and Methods

### 4.1. Plant Material, Growth Conditions, and Sampling

*Rhyncholaeliocattleya* Beauty Girl KOVA (KOVA), with a strong floral scent, was grown in a Chinese Academic Forestry Greenhouse (Beijing, China) under natural light with daytime and nighttime temperatures of 24–30 °C. According to our previous study [25], the sepals, non-lip petals, and the epichile and hypochile regions of the lip petal in KOVA flowers at D7 (i.e., two days after flowering) and D8 (i.e., ten days after flowering) were collected to be used for the experiments in this study. All materials were frozen in liquid nitrogen immediately after sampling and then stored at −80 °C.

### 4.2. Analysis of the Volatile Organic Compounds Emitted by the Flowers

The volatile aroma compounds emitted by live flowers at D7 and D8 were identified using the headspace solid-phase microextraction (SPME) method as previously described [41]. Briefly, live flowers were sheathed with an oven bag for one hour simultaneously, and then the volatile aroma compounds in the bag were collected with a clean 85 μm CAR/PDMS SPME fiber (57334-U, Supelco-Sigma–Aldrich, St Louis, MO, USA) and inserted into a vial to determine the headspace volatiles at one-hour intervals. Finally, the volatile aroma compounds collected in the SPME fiber were analyzed using gas chromatography (GC) (Agilent 7890A, Agilent, Palo Alto, CA, USA). The GC program settings were the same as those described by Zhang et al. [41]. NIST library comparisons and the external standard method with standard chemicals were used to identify and quantify the volatile aroma compounds collected in the SPME fiber.

### 4.3. Identification and Quantification of the Volatile Organic Compounds Produced by Different Parts of Flowers

To identify differences in the contents and types of VOCs from different flower parts, all freeze-dried samples were ground into a powder in liquid nitrogen. Then, 100 mg of the powder was weighed into a headspace bottle and 5 μL of 2-octanol (10 mg/L) was added as an internal standard. Finally, the samples were analyzed using GC–MS (gas chromatograph, Agilent 7890B, Agilent, Palo Alto, CA, USA; mass spectrometer, Agilent 5977B, Agilent, Palo Alto, CA, USA). The SPME program settings of the PAL rail system were as follows: the incubation temperature was 60 °C, the preheating time was 15 min, the incubation time was 30 min, and the desorption time was 4 min. A DB-Wax column (30 m × 250 μm × 0.25 μm) and the splitless injection mode were selected. Helium was used as the carrier gas, the front inlet purge flow was 3 mL min^−1^, and the gas flow rate through the column was 1 mL min^−1^. The initial temperature was maintained at 40 °C for 4 min and then raised to 245 °C at a rate of 5 °C min^−1^, which was maintained for 5 min. The injection, transfer line, ion source, and quad temperatures were 250, 250, 230, and 150 °C, respectively. The energy was −70 eV in the electron impact mode. The mass spectrometry data were acquired in the scan mode with an m/z range of 20–500 and a solvent delay of 0 min. Finally, the identification and quantification of each substance in the samples were conducted from the mass spectrometry data using Chroma TOF 4.3X software and the NIST database.

### 4.4. Transcriptome and Heatmap Analyses

Transcriptome sequencing and analysis were described in detail in our previous study [25]. The transcriptome data used in this study were deposited in the NCBI database under SRA accession codes PRJNA559603 and PRJNA559608. Heatmap analysis was used to show the gene expression profiles using Omiscshare tools (http://omicshare.com/tools/), and if there were more than ten homologous transcripts, only ten homologous transcripts were displayed; all of the homologs are shown in the Appendix A due to the limited size of the picture.

### 4.5. Phylogenetic Analysis

A neighbor-joining phylogenetic tree was constructed with bootstrap values estimated from 1000 replicate runs by MEGA 7 [42] to analyze the relationships between RhMTPs and MTPs from other plants. Protein sequence alignment was estimated with Cluster W. Modification of the NJ tree was performed by Evolgenius (http://www.evolgenius.info/evolview/). The accession numbers of the MTP genes from other plants are as follows: (*4S*)-*limonene synthase* (AAC37366), (*R*)-*limonene synthase* (AAM53944), *QH5* (AAF13356), *OCS* (AGB14628.1), *FhLin* (AFP23421.1), *SlTPS38* (AEP82768.1), *PT1* (AAO61225.1), *FES* (AAV63790.1), *VvaTerp* (NP_001268216.1), and *TPS-CIN* (NM_113483.5).

## Figures and Tables

**Figure 1 plants-12-00304-f001:**
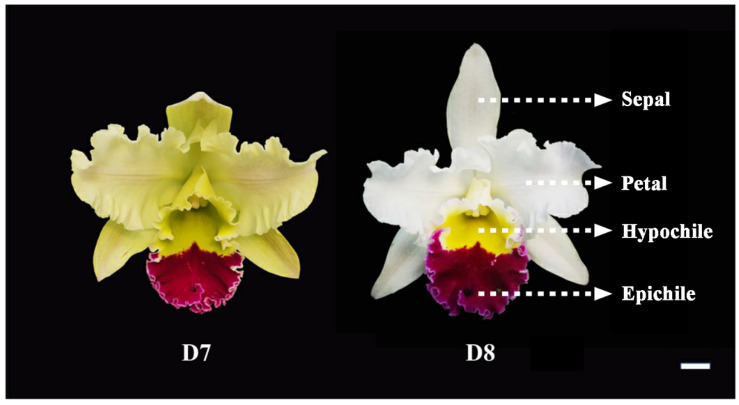
KOVA flowers during flowering. According to a previous study [25], KOVA flowers at developmental stages 7 (D7) and 8 (D8). The structure of a KOVA flower (**right picture**). The KOVA flower is scentless and fragrant at D7 and D8, respectively. Bar = 1 cm.

**Figure 2 plants-12-00304-f002:**
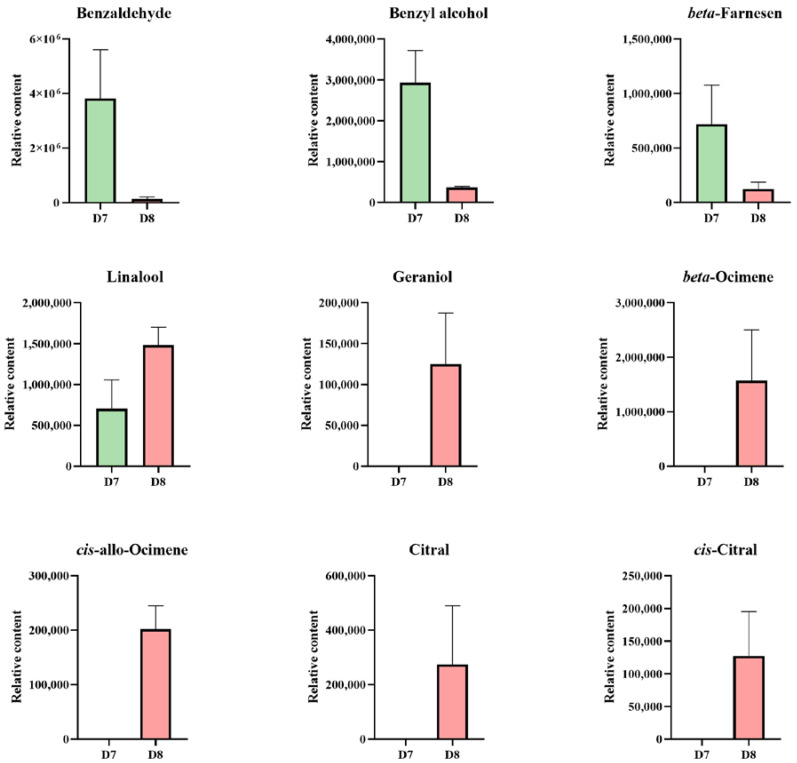
The relative contents of VOCs emitted by KOVA perianths during flowering. The relative content represents the result calculated by the external standard method based on data produced by GC. The data are the mean ± SD from three biological replicates.

**Figure 3 plants-12-00304-f003:**
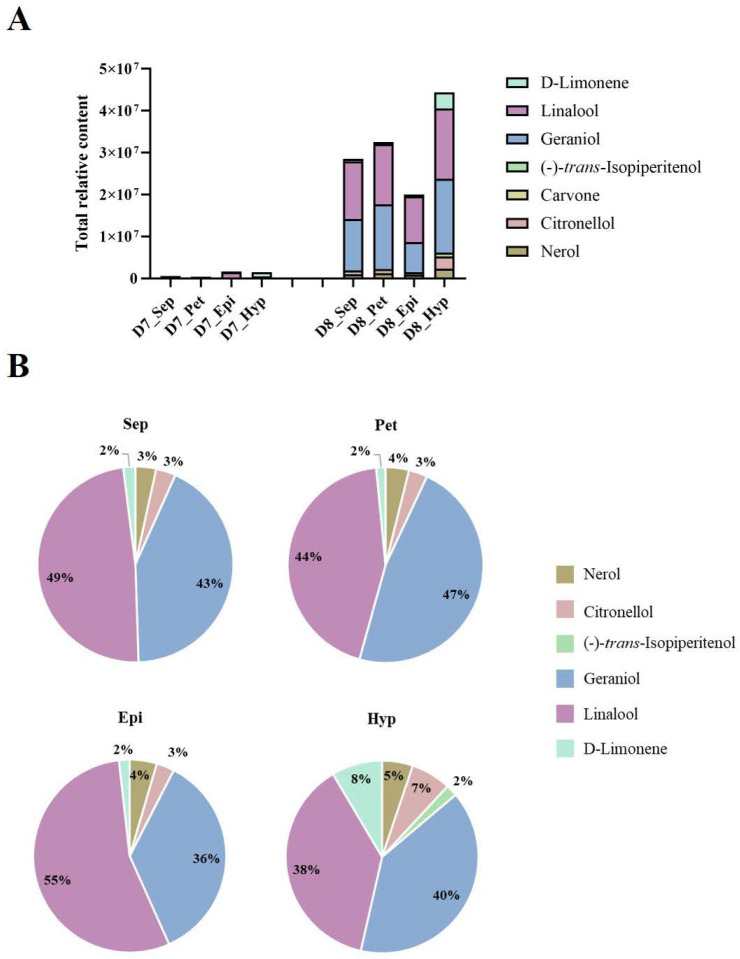
The types and relative contents of monoterpenes in four parts of the KOVA flower at stages D7 and D8. (**A**) The total relative contents of the monoterpenes in each part of the flower at D7 and D8. The four parts of KOVA flowers at D7 and D8 were sampled and analyzed using headspace GC–MS. The sepal, petal, epichile, and hypochile are abbreviated as Sep, Pet, Epi, and Hyp, respectively. The relative content represents the result calculated by the external standard method based on data produced by GC. The total relative content represents the sum total of all monoterpene relative contents. The data are the mean ± SD from three biological replicates. (**B**) The proportion of the relative contents of each monoterpene in the total relative content in different parts of the KOVA flower at D8.

**Figure 4 plants-12-00304-f004:**
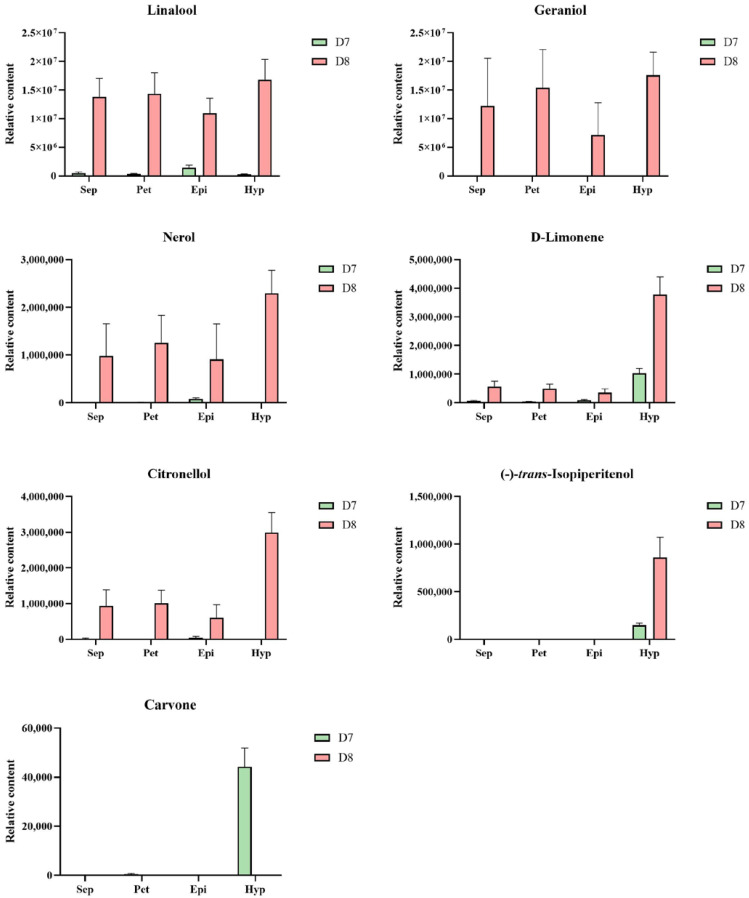
Spatiotemporal accumulation of monoterpenes in KOVA flowers at different developmental stages. The relative contents of each monoterpene in the flower parts at D7 and D8. The relative content represents the result calculated by the external standard method based on data produced by GC. The data are the mean ± SD from three biological replicates.

**Figure 5 plants-12-00304-f005:**
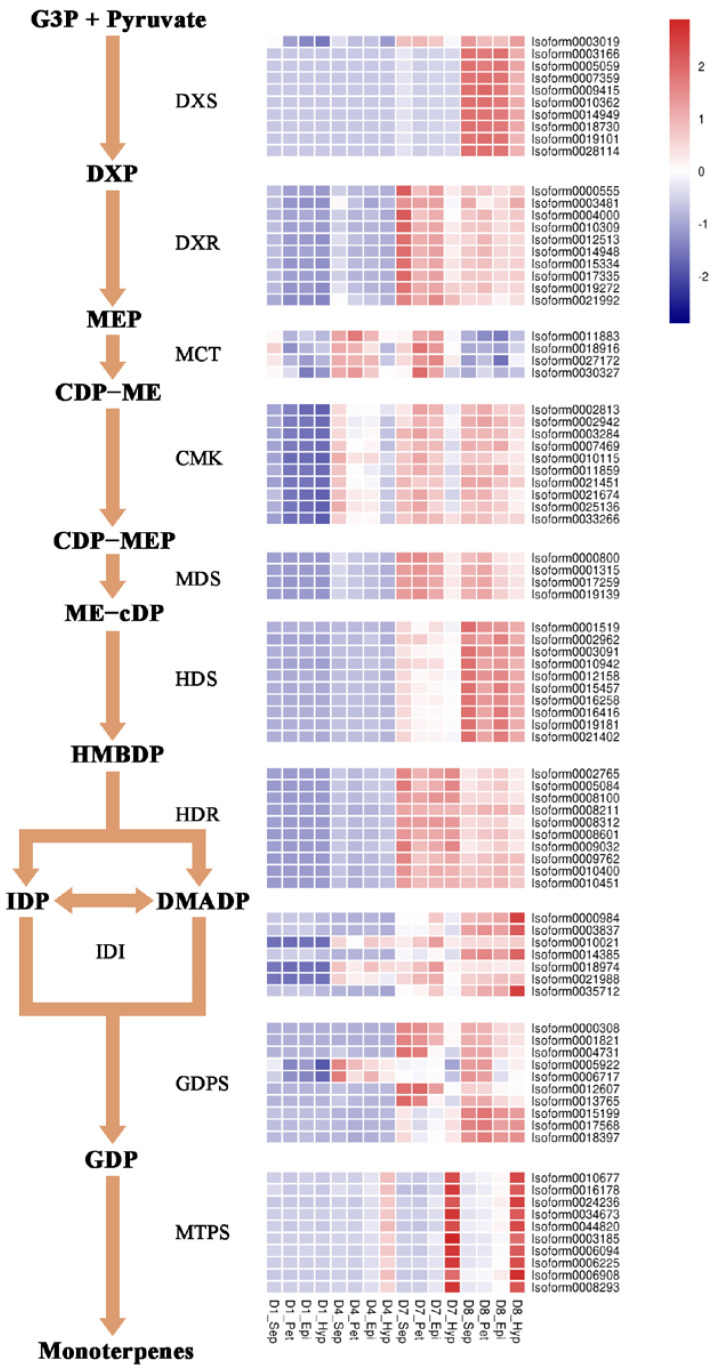
The spatiotemporal expression profiles of MEP pathway structural genes. According to a previous study, KOVA flower development is classified into eight stages, including D1 (i.e., bub length <2 cm), D4 (i.e., bub length 4–5 cm), D7 (i.e., two days after flowering), and D8 (i.e., ten days after flowering) [25]. A heatmap analysis of the expression levels of MEP pathway structural genes in the sepals, petals, epichile, and hypochile from D1 (developmental stage 1) to D8 is displayed. The transcriptome data were provided by a previous study [25] and reanalyzed in this study. If there are more than ten homologous transcripts, only ten homologous transcripts are displayed; the expressions of all MEP pathway genes are displayed in the Appendix A. The data are the mean ± SD from three biological replicates. Abbreviations for compounds or enzymes not described in this article are as follows: G3P, glyceraldehyde-3-phosphate; DXP, 1-deoxy-d-xylulose 5-phosphate; MEP, methylerythritol phosphate; CDP-ME, 4-diphosphocytidyl-2-C-methylerythritol; CDP-MEP, 4-diphosphocytidyl-2-C-methyl-d-erythritol 2-phosphate; ME-cDP, 2-C-methyl-d-erythritol 2,4-cyclodiphosphate; HMBDP, 4-hydroxy-3-methyl-but-2-enyl pyrophosphate; IDP, isopentenyl diphosphate; DMADP, dimethylallyl diphosphate; GDP, geranyl diphosphate.

**Figure 6 plants-12-00304-f006:**
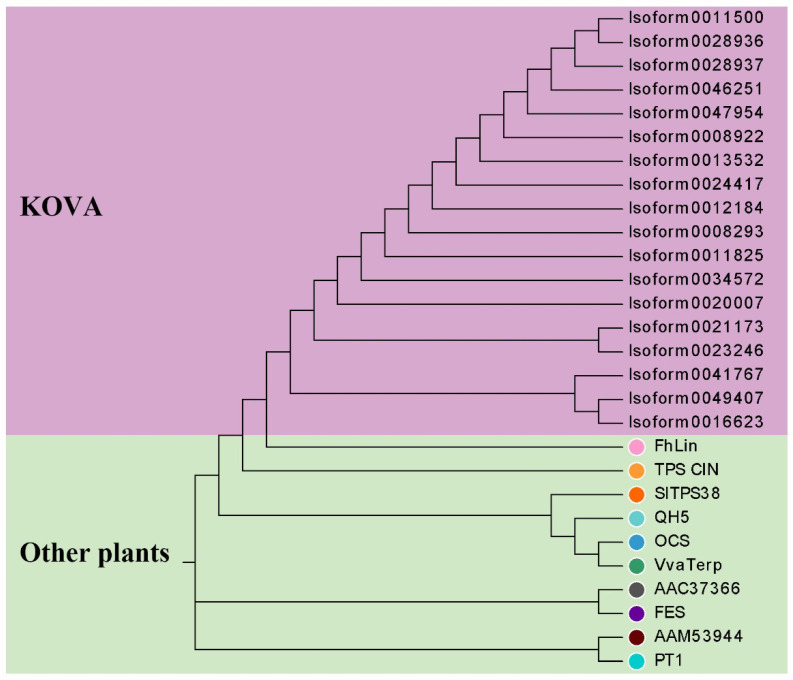
Phylogenetic relationships between RcMTPSs and MTPSs from other plants. All amino acid sequences without repeats from RcMTPs from KOVA and MTPs from other plants were used to establish a phylogenetic tree based on the neighbor-joining method. These data were employed to analyze the relationship between the MTPs. Protein abbreviations are as follows: FhLin, *Freesia hybrid* linalool synthase; TPS-CIN, *Arabidopsis thaliana* terpene synthase-like sequence-1,8-cineole; SlTPS38, *Solanum lycopersicum* terpene synthase 38; QH5, (3R)-linalool synthase; OCS, *trans*-beta-Ocimene synthase; VvaTerp, *Vitis vinifera* (-)-alpha-terpineol synthase; AAC37366, 4S-limonene synthase; FES, fenchol synthase; AAM53944, (R)-limonene synthase; PT1, *Pinus taeda* (-)-alpha-pinene synthase.

## Data Availability

The original contributions presented in the study are included in the article/Appendix A; further inquiries can be directed to the corresponding author.

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
