# Peer review of "New Insights into the Mechanism of Spatiotemporal Scent Accumulation in Orchid Flowers"

_plants, 2023, doi:10.3390/plants12020304_

Round 1

Reviewer 1 Report

The new findings claimed by the authors are as follows.

      At the developmental stage 7 (D7), benzenoids and β-farnesene were the major scent components, and at the developmental stage 8 (D8), monoterpenes were the major scent components. This indicates that the scent of Cattleya hybrid KOVA flowers changes greatly during the transition from D7 to D8.

      At D8, when the monoterpenes were emitted, the scent was felt and thrips visited the flowers. However, D7 was scentless. This means that monoterpenes are the fragrant-contributing component and the attractant for insects in this flower.

      Monoterpene content levels were highest in hypochile, and thrips visits were also concentrated here. This means that hypochile is the primary source of scent.

      The emission of monoterpenes and the expression of 1-deoxy-D-xylulose-5-phosphate synthase (DXS) genes were synchronized. This suggests that the expression of DXS genes regulates monoterpene emissions at D8.

      The expression levels of monoterpene synthase (MTPS) genes were highest in hypochile at D8. This suggests that the expression of MTPS genes contributes to the high emission of monoterpenes at hypochile. 

These findings are novel, and it is particularly noteworthy that both scent compounds and gene expression analyzes suggest that a specific tepals, hypochile, plays an important role in scent emission.

However, the authors' claims are not fully supported by the data. Many imperfections need to be clarified. The reason is as follows.

In this article, the D7 scent is described as "scentless". However, at D7, some benzenoids and sesquiterpenes have been detected (Fig. 1). The olfactory thresholds of these scent compounds are not low, so the data and conclusions appear to be inconsistent. 

The authors interpreted hypochile to have the highest monoterpene content level and speculated that linalool and geraniol, which were the major scent components, were the main factors in the fragrance of KOVA flowers (Fig. 2b). The authors also claimed that hypochile was the most fragrant part of the flower. These conclusions may be correct. However, it appears to this reviewer that there is no remarkable difference between hypochile and other tepals (Fig. 2b). In particular, geraniol and linalool do not seem to differ significantly between tepals (Fig. 2 c). Therefore, the results of Figure 2 cannot reasonably explain the strong scent or the concentration of thrips in hypochile. On the other hand, several minor terpenoids appear hypochile-specific (Fig. 2 c). In any case, deduction of hypochile fragrance factors and thrips attraction factors requires further investigation.

High expression of MTPS genes was shown only in hypochile (Figure 3 and supplementary Figure 2). However, as mentioned above, the monoterpene levels of hypochile and other tepals do not show such a clear difference. Therefore, it remains unclear whether MTPS genes expression is a determinant of monoterpene content in tepals. 

A major cause of these non-negligible problems is the lack of quantification of the actual emission and content of scent components. This reviewer believes that determining the actual amount of scent compounds leads to definite conclusions. It is recommended to also identify the amount of scent emitted from each tepal. 

The KOVA floral were scentless at D7 (L89). The hypochile is more fragrant than the other tissues (L109-110). Perhaps those views are correct. However, whether or not the flowers are fragrant is the subjective opinion of the authors, not an objective opinion based on sensory evaluation. Therefore, this reviewer objects to including such the opinion in the Results. The authors views should be understated in the Discussion. Thrips behavior could be a unique scent phenotype for hypochiles. This reviewer is strongly encouraged to quantify data on thrips behavior.

Other comments

I think it would be better to unify the chemical notation.

For example, unify to either “(E)-” or “trans-”.   “β-” or “beta-”.

* Underlined parts are italic.

Introduction

L25: Flower scents can help plants attract pollinators and are an important commodity for ornamental flowers [1].

This reference [1] does not seem appropriate. References should be made to the literature that describes the results that support the authors' idea (underline).

L54-55: MTPSs are usually regarded as determining the types and contents of monoterpene in flowers [11,15].

One [11] of these references does not seem to claim the underlined view. Also, another reference [15] alone cannot claim to be "usually".

L58: The old Nomenclature convention allowed the use of "cv.", but no longer.

Dianthus caryophyllus cv. Eilat    Dianthus caryophyllus 'Eilat'

L60: PbTPS10 and PbTPS5 should be written in italic.

L64: The scientific name of the thrips must be indicated.

Results

L113: biosynthesized detected”

L114: “produced” → “detected”

L128: 2.3. Expression profiles of the genes involved in the MEP pathway in the KOVA flower parts during floral development

The presence of supplement data should be indicated.

L162-171: A phylogenetic tree ~ KOVA flowers (Fig. 1c; Fig. 2).

This reviewer does not feel strongly the need for the results of this incomplete analysis.

Discussion

The authors showed for the first time the scent components emitted from KOVA flowers. It is recommended to compare the scent components of KOVA and other Rhyncholaeliocattleya flowers. By doing so, the fragrance characteristics of KOVA flowers will be more defined.

L187-192: Among VOCs ~ KOVA flowers.

The authors' view is probably correct, but they fail to explain why the benzenoid and sesquiterpene emissions are 'scentless'.

L192-195: Additionally, ~ other parts of the flower.

This reviewer cannot agree with the authors' view, as there does not appear to be any significant difference in the levels of the major scent compounds such as geraniol and linalool (Fig. 2c).

L196-198: There was little difference ~ methods used.

This reviewer could not understand the intent of the authors in this part.

L201: many thrips usually gather there (Fig. 1b).

In the photograph of Figure 1, thrips also appear to be present on tepals other than the yellow hypochile. The authors need to show that the hypochile has a significantly higher number of thrips. In addition, the effect of tepal color differences on thrips behavior should also be examined.

L201-203: This suggests ~ in the wild,

This reviewer cannot agree with the authors' view, as there does not appear to be any significant difference in the levels of the major scent compounds such as geraniol and linalool (Fig. 2c).

L203-204: although the thrip is not a pollinator for orchids.

Please indicate the reference.

L206: their (= MEP pathway structural genes) expression level can usually influence floral scent directly by affecting terpene accumulation [11].

Does this reference [11] give such a view? 

L207-208: DXS, as the rate-limiting enzyme in the MEP pathway, usually controls the total content of terpenes in plants [30,31].

Of the two references, one [31] is a paper on bacterial metabolism and is not suitable as a citation.

L222-225: These results indicated ~ KOVA flower.

This reviewer considers this view to be premature, as the monoterpene content level and the MTPS gene expression level in each tepal do not correlate (Figures 2c and 3).

L226-227: the high expression of RcDXSs ~ full flowering.

Since it is a guess, it is better to keep the tone low. RcDXS should be written in italic.

The expression pattern of 4-hydroxy-3-methylbut-2-enyl diphosphate synthase (HDS) genes also seems to be similar to that of DXS genes (Fig. 3 and Supplementary Fig. 1). This reviewer think it is necessary to explain why the authors focused only on the DXS gene.

L227-230: The spatiotemporal expression ~ facilitate attracting pollinators (Fig. 5).

The reviewer considers this view to be premature, as the monoterpene content level and the MTP gene expression level in each tepal do not correlate (Figures 2c and 3).

Materials and Methods

The authors claim that the scent of hypochile differs from that of other perianths. Nevertheless, there appears to be no clear difference in the monoterpene content of each tepal (Fig. 2c). As a reason, there may be a problem with the method of collecting scents. For example, the authors may need to consider the following.

It has been shown that there is a circadian rhythm in the scent emission of Rhyncholaeliocattleya flowers (Ren et al., 2015, Chemistry of Natural Compounds). Therefore, at least, all samples should be collected at the same time to reduce the effects of this rhythm.

Collection time with SPME may be too long. Therefore, the amount of aromatic components in the fiber is almost saturated, and there is a possibility that no difference has appeared.

L248: Analysis of the Volatile Organic Compounds emitted from the Flowers

There is no explanation about the method of heat map analysis.

L252 and L258: “extracted” → “collected” ?

L280: 4.3. Transcriptome and heatmap Analyses

It is necessary to explain why the genes shown in Fig. 3 were selected from among many genes.

Figures

Figure 1

In the photograph of Figure 1, thrips also appear to be present on tepals other than the hypochile.

L77: The alphabet of “panel a” in Figure 1 legend is capitalized.

L78: The “(left picture)” in the legend in Figure 1 is unnecessary.

Figure 2

The authors need to explain what the “relative content” is based on. It is difficult to distinguish between low values and non-detection. Please show it so that the difference is understood.

Figure 5

This conclusion is premature.

Author Response

Response to the reviewers’ comments

Reviewer 1:

General comment: The new findings claimed by the authors are as follows.

・      At the developmental stage 7 (D7), benzenoids and β-farnesene were the major scent components, and at the developmental stage 8 (D8), monoterpenes were the major scent components. This indicates that the scent of Cattleya hybrid KOVA flowers changes greatly during the transition from D7 to D8.

・      At D8, when the monoterpenes were emitted, the scent was felt and thrips visited the flowers. However, D7 was scentless. This means that monoterpenes are the fragrant-contributing component and the attractant for insects in this flower.

・      Monoterpene content levels were highest in hypochile, and thrips visits were also concentrated here. This means that hypochile is the primary source of scent.

・      The emission of monoterpenes and the expression of 1-deoxy-D-xylulose-5-phosphate synthase (DXS) genes were synchronized. This suggests that the expression of DXS genes regulates monoterpene emissions at D8.

・      The expression levels of monoterpene synthase (MTPS) genes were highest in hypochile at D8. This suggests that the expression of MTPS genes contributes to the high emission of monoterpenes at hypochile. 

These findings are novel, and it is particularly noteworthy that both scent compounds and gene expression analyzes suggest that a specific tepals, hypochile, plays an important role in scent emission.

However, the authors' claims are not fully supported by the data. Many imperfections need to be clarified. The reason is as follows.

Response: Thank you for your patience and suggestions. We have improved the manuscript. The major revised portions are indicated with track changes in this revised manuscript.

Comment 1: In this article, the D7 scent is described as "scentless". However, at D7, some benzenoids and sesquiterpenes have been detected (Fig. 1). The olfactory thresholds of these scent compounds are not low, so the data and conclusions appear to be inconsistent.

Response: Thank you very much. We have changed Figure 1 and relative description in this manuscript. Please see Figure 2 and line 79. (in red)

Comment 2: The authors interpreted hypochile to have the highest monoterpene content level and speculated that linalool and geraniol, which were the major scent components, were the main factors in the fragrance of KOVA flowers (Fig. 2b). The authors also claimed that hypochile was the most fragrant part of the flower. These conclusions may be correct. However, it appears to this reviewer that there is no remarkable difference between hypochile and other tepals (Fig. 2b). In particular, geraniol and linalool do not seem to differ significantly between tepals (Fig. 2 c). Therefore, the results of Figure 2 cannot reasonably explain the strong scent or the concentration of thrips in hypochile. On the other hand, several minor terpenoids appear hypochile-specific (Fig. 2 c). In any case, deduction of hypochile fragrance factors and thrips attraction factors requires further investigation.

Response: We have reconstructed Figure 2 and named as Figure 3, and we have deleted anything about the thrips in this manuscript. Additionally, we agree with the reviewer’s view that other minor terpenoids appear hypochile-specific and have rewritten this parts in Results and Discussion section. (in red)

Comment 3: High expression of MTPS genes was shown only in hypochile (Figure 3 and supplementary Figure 2). However, as mentioned above, the monoterpene levels of hypochile and other tepals do not show such a clear difference. Therefore, it remains unclear whether MTPS genes expression is a determinant of monoterpene content in tepals.

Response: The total relative content of monoterpene was the highest in hypochile that was in accordance with the expression profiles of MTPSs, which suggested that the highest expression level of MTPs in hypochile might contribute to the highest total relative content of monoterpene in the hypochile. We have made speculation carefully in this manuscript. Please see Line 246-269. (in red)

Comment 4: A major cause of these non-negligible problems is the lack of quantification of the actual emission and content of scent components. This reviewer believes that determining the actual amount of scent compounds leads to definite conclusions. It is recommended to also identify the amount of scent emitted from each tepal.

Response: We very agree with the reviewer’s suggestion that identify the amount of scent emitted from each tepal. However, this is difficult to collect the VOCs from each tepal due to complicated flower structure.

Comment 5: The KOVA floral were scentless at D7 (L89). The hypochile is more fragrant than the other tissues (L109-110). Perhaps those views are correct. However, whether or not the flowers are fragrant is the subjective opinion of the authors, not an objective opinion based on sensory evaluation. Therefore, this reviewer objects to including such the opinion in the Results. The authors views should be understated in the Discussion. Thrips behavior could be a unique scent phenotype for hypochiles. This reviewer is strongly encouraged to quantify data on thrips behavior.

Response: We have made a statement that the presence of scent in different floral parts of KOVA is strictly based on perception by the human nose. Please see Line 79. (in red)

Comment 6: I think it would be better to unify the chemical notation. For example, unify to either “(E)-” or “trans-”.   “β-” or “beta-”. * Underlined parts are italic.

Response: We have revised in this manuscript accordingly to the reviewer’s suggestion. (in red)

Comment 7: L25: Flower scents can help plants attract pollinators and are an important commodity for ornamental flowers [1]. This reference [1] does not seem appropriate. References should be made to the literature that describes the results that support the authors' idea (underline).

Response: We have revised that and added some appropriate references. Please see Line 31-32. (in red)

Comment 8: L54-55: MTPSs are usually regarded as determining the types and contents of monoterpene in flowers [11,15]. One [11] of these references does not seem to claim the underlined view. Also, another reference [15] alone cannot claim to be "usually".

Response: We have revised that in this manuscript. Please see Line 58-59. (in red)

Comment 9: L58: The old Nomenclature convention allowed the use of "cv.", but no longer. Dianthus caryophyllus cv. Eilat  →  Dianthus caryophyllus 'Eilat'

Response: We have revised that in this manuscript. (in red)

Comment 10: L60: PbTPS10 and PbTPS5 should be written in italic.

Response: We have revised that in this manuscript. (in red)

Comment 11: L64: The scientific name of the thrips must be indicated.

Response: We have deleted anything about the thrips in this manuscript.

Comment 12: L113: “biosynthesized” → “detected” L114: “produced” → “detected”

Response: We have revised that in this manuscript. (in red)

Comment 13: L128: 2.3. Expression profiles of the genes involved in the MEP pathway in the KOVA flower parts during floral development. The presence of supplement data should be indicated.

Response: We have revised that in supplement data. Please see supplement data

Comment 14: L162-171: A phylogenetic tree ~ KOVA flowers (Fig. 1c; Fig. 2).

This reviewer does not feel strongly the need for the results of this incomplete analysis.

Response: We think that it can provide useful information for this study and readers, we therefore remain this result.

Comment 15: The authors showed for the first time the scent components emitted from KOVA flowers. It is recommended to compare the scent components of KOVA and other Rhyncholaeliocattleya flowers. By doing so, the fragrance characteristics of KOVA flowers will be more defined.

Response: Thank your precious suggestion, we have rewritten this part in Discussion section. (in red)

Comment 16: L187-192: Among VOCs ~ KOVA flowers.

The authors' view is probably correct, but they fail to explain why the benzenoid and sesquiterpene emissions are 'scentless'.

Response: We have rewritten this part in Discussion section, please see Line 218-245. (in red)

Comment 17: L192-195: Additionally, ~ other parts of the flower.

This reviewer cannot agree with the authors' view, as there does not appear to be any significant difference in the levels of the major scent compounds such as geraniol and linalool (Fig. 2c).

Response: We have redisplayed Figure 2c to name as Figure 2, and combing it we discussed this part in Discussion section, please see 222-236. (in red)

Comment 18: L196-198: There was little difference ~ methods used.

This reviewer could not understand the intent of the authors in this part.

Response: We have deleted this sentence. (in red)

Comment 19: L201: many thrips usually gather there (Fig. 1b).

In the photograph of Figure 1, thrips also appear to be present on tepals other than the yellow hypochile. The authors need to show that the hypochile has a significantly higher number of thrips. In addition, the effect of tepal color differences on thrips behavior should also be examined.

Response: We have deleted anything about the thrips in this manuscript. (in red)

Comment 20: L201-203: This suggests ~ in the wild,

This reviewer cannot agree with the authors' view, as there does not appear to be any significant difference in the levels of the major scent compounds such as geraniol and linalool (Fig. 2c).

Response: We have rewritten this part in Discussion section, please see Line 218-245. (in red)

Comment 21: L203-204: although the thrip is not a pollinator for orchids.

Please indicate the reference.

Response: We have deleted anything about the thrips in this manuscript. (in red)

Comment 22: L206: their (= MEP pathway structural genes) expression level can usually influence floral scent directly by affecting terpene accumulation [11].

Does this reference [11] give such a view? 

Response: We have rewritten this sentence, please see line 246-247. (in red)

Comment 23: L207-208: DXS, as the rate-limiting enzyme in the MEP pathway, usually controls the total content of terpenes in plants [30,31].

Of the two references, one [31] is a paper on bacterial metabolism and is not suitable as a citation.

Response: We have deleted this reference. (in red)

Comment 24: L222-225: These results indicated ~ KOVA flower.

This reviewer considers this view to be premature, as the monoterpene content level and the MTPS gene expression level in each tepal do not correlate (Figures 2c and 3).

Response: We have redisplayed Figure 2c and 3 to Figure 2, 3, and 4, please see these figures and we have remade appropriate discussion, please see 246-269. (in red)

Comment 25: L226-227: the high expression of RcDXSs ~ full flowering.

Since it is a guess, it is better to keep the tone low. RcDXS should be written in italic.

The expression pattern of 4-hydroxy-3-methylbut-2-enyl diphosphate synthase (HDS) genes also seems to be similar to that of DXS genes (Fig. 3 and Supplementary Fig. 1). This reviewer think it is necessary to explain why the authors focused only on the DXS gene.

Response: Thank your precious suggestion, we have rewritten this part in Discussion section. Please see line 246-269. (in red)

Comment 26: L227-230: The spatiotemporal expression ~ facilitate attracting pollinators (Fig. 5).

The reviewer considers this view to be premature, as the monoterpene content level and the MTP gene expression level in each tepal do not correlate (Figures 2c and 3).

Response: We have deleted Figure 5 and relative sentence about pollination. (in red)

Comment 27: The authors claim that the scent of hypochile differs from that of other perianths. Nevertheless, there appears to be no clear difference in the monoterpene content of each tepal (Fig. 2c). As a reason, there may be a problem with the method of collecting scents. For example, the authors may need to consider the following. It has been shown that there is a circadian rhythm in the scent emission of Rhyncholaeliocattleya flowers (Ren et al., 2015, Chemistry of Natural Compounds). Therefore, at least, all samples should be collected at the same time to reduce the effects of this rhythm. Collection time with SPME may be too long. Therefore, the amount of aromatic components in the fiber is almost saturated, and there is a possibility that no difference has appeared.

Response: Thank for your suggestion. We collected VOCs emitting from KOVA flower at D7 and D8 simultaneously, and this method of collection time was performed accordingly to Zhang et al (Zhang, W.; Liu, L.; He, S.; Lu, B.; Luo, Y. The production and evolution pattern of “fruity smell" aggregation pheromones in genus Drosophila. J. Sys. Evol. 2020, 60, 208–219; DOI: https://doi.org/10.1111/jse.12648).

Comment 28: L248: Analysis of the Volatile Organic Compounds emitted from the Flowers

There is no explanation about the method of heat map analysis.

Response: We have change the Figure 2c to Figure 2 and not used the heat map analysis. (in red)

Comment 29: L252 and L258: “extracted” → “collected” ?

Response: We have revised that in this manuscript. (in red)

Comment 30: L280: 4.3. Transcriptome and heatmap Analyses

It is necessary to explain why the genes shown in Fig. 3 were selected from among many genes.

Response: We have added this part in this section. (in red)

Comment 31: Figure 1

In the photograph of Figure 1, thrips also appear to be present on tepals other than the hypochile.

Response: We have deleted anything about thrips in this manuscript and redisplayed Figure 1. Please see Figure 1. (in red)

Comment 32: L77: The alphabet of “panel a” in Figure 1 legend is capitalized.

L78: The “(left picture)” in the legend in Figure 1 is unnecessary.

Response: We have redisplayed Figure 1 and revised that in the legend. (in red)

Comment 33: Figure 2

The authors need to explain what the “relative content” is based on. It is difficult to distinguish between low values and non-detection. Please show it so that the difference is understood.

Response: We have redisplayed figure 2 to figure 2 and 3. Please see Figure 2 and 3. (in red)

Comment 34: Figure 5

This conclusion is premature.

Response: We have deleted that. (in red)

Reviewer 2 Report

Dear authors,

I am grateful for the opportunity to review your manuscript as a peer, which consider interesting and well accomplished. 

A few comments about it.

It is mentioned in the text, regarding the relative concentration of higher monoterpenes in the hypochile. However, and given that the use of standards is described in Materials and Methods, I believe that it would be interesting (at least as supplementary material) to have the chromatograms and their results in tables in view, in relation to the use of internal standards or external, in order to show real and not relative or percentage concentrations. The fact of mentioning that Carvone is only produced in the hypochile seems somewhat exhaustive if it is also considered that there are possibly traces of this metabolite in other floral parts or that it is found in fresh parts of the flower. (lines 113 to 116). This last assessment is not consistent with what is described in lines 193 and 194.

In line 241, it seems more appropriate to indicate that the species under study has a floral essence, not necessarily fascinating.

As I mentioned before, in accordance with what was described in line 258, it would be interesting, I reiterate, to have in view the results of the quantification by comparison via external standard.

Finally, it should be noted that reference 19 seems to have been included after the preparation of the text, since its appearance in the text is not continuous, going from 18 to 20 and only a couple of pages later it refers to 19. If possible, it would be good to modify for better understanding and reading.

Author Response

Response to the reviewers’ comments

Reviewer 2:

General comment: I am grateful for the opportunity to review your manuscript as a peer, which consider interesting and well accomplished. 

Response: Thank you for your patience and suggestions. We have improved the manuscript. The major revised portions are indicated with track changes in this revised manuscript.

Comment 1: It is mentioned in the text, regarding the relative concentration of higher monoterpenes in the hypochile. However, and given that the use of standards is described in Materials and Methods, I believe that it would be interesting (at least as supplementary material) to have the chromatograms and their results in tables in view, in relation to the use of internal standards or external, in order to show real and not relative or percentage concentrations. The fact of mentioning that Carvone is only produced in the hypochile seems somewhat exhaustive if it is also considered that there are possibly traces of this metabolite in other floral parts or that it is found in fresh parts of the flower. (lines 113 to 116). This last assessment is not consistent with what is described in lines 193 and 194.

Response: Thank for your precious suggestions. We have redisplayed Figure 1 and 2 to Figure 1, 2, 3, please see those. Additionally, we have revised that accordingly to your suggestion that ‘The fact of mentioning that Carvone is only produced in the hypochile seems somewhat exhaustive if it is also considered that there are possibly traces of this metabolite in other floral parts or that it is found in fresh parts of the flower. (lines 113 to 116). This last assessment is not consistent with what is described in lines 193 and 194.’, please see Line 143-145 and Line 231-233. (in red)

Comment 2: In line 241, it seems more appropriate to indicate that the species under study has a floral essence, not necessarily fascinating.

Response: We have revised that, please see line 241. (in red)

Comment 3: As I mentioned before, in accordance with what was described in line 258, it would be interesting, I reiterate, to have in view the results of the quantification by comparison via external standard.

Response: Thank for your precious suggestions. We made quantification of VOCs with a relative content, which also could display the changes of contents of VOCs emitting or producing from KOVA flower and tissues, respectively.

Comment 4: Finally, it should be noted that reference 19 seems to have been included after the preparation of the text, since its appearance in the text is not continuous, going from 18 to 20 and only a couple of pages later it refers to 19. If possible, it would be good to modify for better understanding and reading.

Response: Thank for your precious reminders, we have revised that in this manuscript. (in red)

Author Response

Response to the reviewers’ comments

Reviewer 3:

General comment: This manuscript reports on interesting findings on the scent chemistry of the orchid Cattleya hybrid KOVA, the location of its emission among the flower parts, and the location of enzymes involved in the biosynthesis of the main floral volatiles. The reading was overall smooth, but there are some problems that need addressing, including omitting redundancy (there are several phrases that are repeated many times in the manuscript), reviewing the literature on orchid scents much more rigorously while also removing the random citations to irrelevant papers on floral scents that address totally unrelated plant species, backing up various comments with citations of the literature, and in the discussion placing the findings of this orchid within the context of what we know about other orchids’ intra-floral scent patterns and how they influence pollination.

Response: Thank you for your patience and suggestions. We have improved the manuscript. The major revised portions are indicated with track changes in this revised manuscript.

Comment 1: Here and everywhere in the manuscript, the authors need to clarify the flower parts of orchids, which in fact have 3 sepals and 3 petals, with one of the petals forming the labellum (lip). The lip can be differentiated into the hypochile and epichile. In the manuscript the lip is treated like a separate flower part (it is one of the petals!), and they regions of the lip are not properly introduced before they are referred to in the text.

Response: We have revised that in the whole manuscript. (in red)

Comment 2: Line 8: the lip is one of the petals

Response: We have revised that in the whole manuscript. Please see line 11. (in red)

Comment 3: Line 9: replace “complicated” with “complex”; replace “conflicting” with “differing”; what do you mean by “mechanisms? This is too vague – do you mean factors regulating scent production, or?

Response: We have revised that in the manuscript. Please see line 13. (in red)

Comment 4: Line 10: omit the word “mechanism”

Response: We have deleted that.

Comment 5: Line 11: put a comma after “monoterpenes”; add “which” to …”geraniol, which are the…

Response: We have revised that in the manuscript. Please see line 16. (in red)

Comment 6: Line 12: How can you state that it was scentless when volatiles were emitted? Specify that you mean “scentless to the human nose

Response: Yes, we mean “scentless to the human nose” and have revised that in the whole manuscript. Please see line 17. (in red)

Comment 7: Line 13: replace “completely flowered” to “reached full bloom”

Response: We have revised that in the manuscript. Please see line 18. (in red)

Comment 8: Line 14: replace “after full flowering” to “when it reached full flowering”

Response: We have revised that in the manuscript. Please see line 21. (in red)

Comment 9: Line 16: define what hypochile is or just say” basal region of the lip”

Response: We have revised that in the manuscript. Please see line 23. (in red)

Comment 10: Line18: omit KOVA

Response: We have deleted that.

Comment 11: Line 18-19: the last sentence is a little empty and needs more substance (the same applies to the end of the discussion)

Response: We have revised that in the manuscript. Please see line 25-27. (in red)

Comment 12: Line 25: “biochemical mechanisms”

Response: We have revised that in the manuscript. Please see line 32. (in red)

Comment 13: Line 27: be more precise: “Orchids… famous families of plants…” They are NOT unique in having 3 sepals and three petals. But they are unique in that one of the three is differentiated into the lip (labellum) *Give a citation here

Response: We have revised that and added a reference in the manuscript. Please see line 34-37. (in red)

Comment 14: Lines 29-31: The last two sentences are confusing and scientifically not correct. “The composition and intensity of scent volatiles has been shown to differ between the different flower parts (add references), and the underlying biochemistry of volatile synthesis has been investigated in some Phalaenopsis and Dendrobium orchids.

Response: We have revised that in the manuscript. Please see line 37-40. (in red)

Comment 14: Paragraph 2: this addresses the chemical composition of floral scents in orchids, and the literature needs to be more rigorously reviewed and not restricted to the few taxa that have also been studied in terms of biosynthetic pathways. There have been many studies of floral scents in a large number of species, using headspace volatile collection.

Response: We have revised that in the manuscript. Please see line 41-57. (in red)

Comment 15: Lines 32-33: “ Terpenoids,.. scents, are classified…types and consist mainly of sesquiterpenes…derivatives. These groups are classified according to…”

Response: We have revised that in the manuscript. Please see line 41-43. (in red)

Comment 16: Lines 36: Omit “Among these terpenoids” -it is clearer when the writing is less wordy and more concise. Also add “especially”: “…found to be especially important…”

Response: We have revised that in the manuscript. Please see line 46. (in red)

Comment 17: Line 37: cite literature that supports this statement

Response: We have added several references in this statement. Please see line 47. (in red)

Comment 18: Lines 38-42: These come across as a random assortment of citations. Cite reviews of orchid scents (there are several) and list some of the monoterpenes that tend to be most prevalent, and then give a few examples of how different ones are dominant in different species.

Response: We have rewritten that in the manuscript. Please see line 47-57. (in red)

Comment 19: Line 62: give more information about your study species: where is it native? In what habitat does it grow? If it is a horticultural hybrid, say so and give the parental species and information about them. How many species are in the genus Rhyncholaeliocattleya? Then describe floral features that distinguish this genus from other orchid genera

Response: We have added more information about Rhyncholaeliocattleya accordingly to the reviewer’s precious suggestions in the manuscript. Please see line 77-79. (in red)

Comment 20: Lines 63-66: give references to these statements! Or are these not documented and simply what the authors sense? State clearly that your statements on the presence of scent in different floral parts of KOVA is strictly based on perception by the human nose. Also, define the different parts of the lip with precision (basal vs apical). And where does this statement about thrips come from? Is this published or simply observations by the authors (say so: “personal observation”).

Response: We have revised that in the manuscript. Please see line 79-82. (in red)

Comment 21: Line 66-67: this sentence is empty – any flower is suitable material for exploring spatial differences in scent: Omit this sentence.

Response: We have deleted that.

Comment 22: Line 67-70: “In this study, we aim to describe the spatiotemporal patterns of 1) scent emission in KOVA flowers, based on headspace scent collection and GC-MS analysis and 2) biosynthetic pathways for the scent volatiles, using transcriptome analyses. We found that…”

Response: We have revised that in the manuscript. Please see line 82-85. (in red)

Comment 23: Figure 1. o Give only a and b here, and cite in the Introduction. o Put 1c into a new and separate figure, and cite in the Results. o In 1a, label the lip. Perhaps you could put the labels for sepals and petals (including the lip) on the D7 and label only the regions of the lip on the D8

Response: Thanks for your precious suggestion, and we have redisplayed the Figure 1. Please see Figure 1.

Comment 24: Lines 86-88: omit

Response: We have deleted that.

Comment 25: Lines 89: omit statement that the flowers were scentless at D7 when clearly they emitted volatiles!

Response: We have deleted that.

Comment 26: Line 91: what do you mean by flower development? This study is only looking at flowers in D7 and D8 stages, both of which of open flowers, with D8 being most fully open. Just state “…during the D7 stage

Response: We have revised that in the manuscript. Please see line 104. (in red)

Comment 27: Lines 92-98: explain at the start which of the volatiles in KOVA are benzenoids, which are sesquiterpenes, and which are monoterpenes.

Response: We have revised that in the manuscript. Please see line 100-102. (in red)

Comment 28: Figure 1c: this is very problematical: there is a heat map, but only two colors (two intensities) are shown, so this heat map is useless and inappropriate. Maybe turn this into a table, and be sure to label the monoterpenes, since they are the volatiles that you are focusing on.

Response: We have revised Figure 1c to Figure 2 accordingly to the reviewer’s precious suggestion, please see Figure 2.

Comment 29: Line 99: change to “Spatiotemperal quantification of VOCs in KOVA flowers

Response: We have revised that in the manuscript. Please see line 116. (in red)

Comment 30: Line 100-104: delete; just give results of this section. Do not confuse the headspace results with the tissue analysis results. Keep them clearly separate. This first paragraph is jumbled and needs much more clarity and precision.

Response: We have deleted that.

Comment 31: Lines 105-111: In presenting the results, point out clearly differences with the headspace (number of compounds and state which are present in tissues only).

Response: We have revised that in the manuscript. Please see line 120-132 and Figure 2. (in red)

Comment 32: Line 109: omit “ which is consistent… other tissues”. This has been stated already (!)

Response: We have deleted that.

Comment 33: Figure 2: This is very difficult to read o Make fonts larger in all graphs o Put the relative content into % (as stated in the text and much more easy for readers to understand) o 2b: separate spatially the columns of D7 and those of D8

Response: We have revised the Figure 2. Please see Figure 2 and 3. (in red)

Comment 34: Lines 113-115: “Most… were present in…D8, except carvone, which was only …(Fig 2C). …monoterpene was highest in the hypochile.

Response: We have revised that in the manuscript. Please see line 143-145. (in red)

Comment 35: Lines 115-117: omit sentence of fragrant parts… too repetitive

Response: We have deleted that.

Comment 36: Lines 117-119: “The relative contents… isopiperitenol at stage D8 were notably higher in the hypochile than in other parts of the flower

Response: We have revised that in the manuscript. Please see line 146-148. (in red)

Comment 37: Line 131: do you mean reference 19 (you give 20)?

Response: Yes, we have revised this problem in the whole manuscript. (in red)

Comment 38: Line 162: omit “built”

 Line 170: omit “composing the floral scents found”

Response: We have deleted that.

Comment 39: Lines 183-185: Cite here a general review on floral scents, such as Knudsen and Gerschenzon, and give a sense to the reader of how prevalent monoterpenes are across floral scents in general. But why single out this random set of three species? This makes no scientific sense. If you provide examples of specific species, their selection needs to be justified. I suggest that you omit the three examples and instead focus your detailed discussion and examples on the family of orchids and, even more, on the genera and species most closely related to your study species

Response: We have revised that in the manuscript. Please see line 218-221. (in red)

Comment 39: Line 186-187: “… the main floral VOCs emitted by the orchids Phalaenopsis…” Omit “flowers” at the end of the sentence.

Response: We have revised that in the manuscript. Please see line 219-221. (in red)

Comment 40: Lines 187-188: “…flowers, only monoterpenes increased quantitatively from…

Response: We have revised that in the manuscript. Please see line 222-224. (in red)

Comment 41: *Line 190: “showed a tendency to decline”: this statement is not accurate, since this is NOT shown in the heatmap of fig 1c, where they seemed to be either detected in high (max in heat map) amounts or in minimal amounts. Omit “which… stages”.

Response: We have revised that and Figure 2 and 3 in the manuscript. Please see line 226 and Figure 2 and 3. (in red)

Comment 42: Line 194: “Additionally, at the D8 stage of bloom, the contents of several … in the tissues of the hypochile region of the lip than in the other flower parts.” Omit “which is the reason…flower”: Too repetitive

Response: We have revised that in the manuscript. Please see line 222-227. (in red)

Comment 43: Lines 196-199: This is confusing: what two sets of data are you referring to? This needs to be made more clear

Response: We have deleted that.

Comment 44: Line 198: only orchids have a petal modified into a “lip” (labellum): “The lip in orchid flowers acts as… fragrance.”

Response: We have revised that in the manuscript. Please see line 240. (in red)

Comment 45: Line 200: the hypochile is basal – use morphological terminology for greater clarity?

Response: We have revised that in the manuscript accordingly to the reviewer’s suggestion. Please see line 242. (in red)

Comment 46: Line 201: again cite the source of the thrips observation

Response: We have deleted anything about thrips in the whole manuscript.

Comment 47: Line 204: what do we know about the pollinators of the parents of KOVA? Elaborate here.

Response: We do not know about the pollinators of parents of KOVA, and we rediscussed and made a conclusion carefully in the manuscript. Please see line 241-245.

Comment 48: Lines 215-220: Tie these statements together into a smooth synthesis - what is your point in giving these specific examples? As examples of how MTPs can be spatial expressed within flowers? If yes, then say so.

Response: We have revised that in the manuscript accordingly to the reviewer’s suggestion. Please see line 264-265. (in red)

Comment 49: Line 224: omit “which…flower”. Too repetitive. (It should be stated only once in the discussion) And again, remember that fragrant here seems to be based on the human nose, not on headspace analysis

Response: We have revised that in the manuscript. Please see line 267-269. (in red)

Comment 50: Line 227: “.. for the high amounts…during the full flowering stage D8.

Response: We have revised that in the manuscript. Please see line 271. (in red)

Comment 51: Line 229: “… compared to other flower parts, which results… fragrant possibly to pollinators.”

Response: We have revised that in the manuscript. Please see line 273-274. (in red)

Comment 52: Line 230-231: Close the discussion with a more substantive sentence. So how does this study concretely expand our understanding of floral scent biology in orchids and what are the logical follow-up studies needed?

Response: We have revised that in the manuscript. Please see line 274-279. (in red)

Comment 53: Line 244: “…sepals, non-lip petals, and the epichile and hypochile regions of the lip petal in KOVA…

Response: We have revised that in the manuscript. Please see line 286-287. (in red)

Round 2

Reviewer 1 Report

This reviewer gave a reject ' conclusion in the previous review. There is no change in that decision this time. The main reason is that there does not appear to be a significant difference in monoterpene endogenous levels between hypochile and other tepals. The authors' claims are based on correlations between relative values of compounds and gene expression, with no causal results to date. Therefore, consistency of data and claims is very important. However, this issue remains unresolved in this revised manuscript.

This reviewer suggested quantitative determination of scent compounds. Standards of many the detected scent compounds are readily available and can be easily quantified. A more objective and concrete discussion is possible by comparing the amounts of compounds. This reviewer also proposed the use of thrips as an indicator as a means of proving a causal relationship between the presence or absence of scent and gene expression. Although this proposal is not essential to the progress of the research, it is easier than the sensory test of scent using human subjects, and it will enhance the evidence for the presence or absence of scent. The authors constructed a phylogenetic tree of monoterpene synthetic enzyme genes. I believe that a little more digging into this analysis would improve the quality of the paper. For example, the main scent components of KOVA flowers were linalool and geraniol. It may be worth discussing whether KOVA MTPSs have the primary sequence characteristics of known enzymes that preferentially catalyze linalool and geraniol synthesis.

Finally, the focus of this research is unique and novel. It's very interesting research, I expect further development.

Other comments

The authors' efforts have greatly improved the presentation. In particular, this reviewer appreciates the authors' softening of the tone to match the quality of the data.

Results

L249: "Ocimene" is usually used as a generic term. Is this trans-beta-Ocimene?

L250, : cis-allocimene → cis-allo-Ocimene

L336: "lower" → "higher"?

Discussion

L339-342: Some monoterpenes, including nerol, D-limonene, (-)-trans-isopiperitenol, and citronellol, were only detected in KOVA flower tissues while ocimene, cis-allocimene, cis-citral, and citral also were not detected (Fig.2; Fig.3).

The previous manuscript did not make this point, so I could see why I did not understand a part of the previous Discussion. It often seems that the compositions of emitted and endogenous scents do not match ( https://doi.org/10.1271/bbb.70490 and https://doi.org/10.1111/ppl.12849 ). This reviewer recommend that the authors refer to these studies and discuss them in the Discussion.

L521: cis-allocimene → cis-allo-Ocimene

L525: the human nose → our nose

By clarifying the criteria for judging the presence or absence of scent, it will be a more accurate description.

L692: The statement of "spatial discrepancy" contradicts the views (L376-379 and L426-429) and claims (L429-431 or L534-537) of the authors. I think a more detailed explanation is required.

Figure 3

The origin of the "Relative content" should be explained in the figure legend.

"Ocimene" is usually used as a generic term. Is this trans-beta-Ocimene?

Please unify the notation of compounds.

For example,

beta-farnesen beta-Farnesen

cis-allocimene → cis-allo-Ocimene

Figure 4

The origin of the "Total relative content" and "percentage" should be explained in the figure legend.

Author Response

Response to the reviewers’ comments

Reviewer 1:

General comment: This reviewer gave a ‘reject ' conclusion in the previous review. There is no change in that decision this time. The main reason is that there does not appear to be a significant difference in monoterpene endogenous levels between hypochile and other tepals. The authors' claims are based on correlations between relative values of compounds and gene expression, with no causal results to date. Therefore, consistency of data and claims is very important. However, this issue remains unresolved in this revised manuscript.

This reviewer suggested quantitative determination of scent compounds. Standards of many the detected scent compounds are readily available and can be easily quantified. A more objective and concrete discussion is possible by comparing the amounts of compounds. This reviewer also proposed the use of thrips as an indicator as a means of proving a causal relationship between the presence or absence of scent and gene expression. Although this proposal is not essential to the progress of the research, it is easier than the sensory test of scent using human subjects, and it will enhance the evidence for the presence or absence of scent. The authors constructed a phylogenetic tree of monoterpene synthetic enzyme genes. I believe that a little more digging into this analysis would improve the quality of the paper. For example, the main scent components of KOVA flowers were linalool and geraniol. It may be worth discussing whether KOVA MTPSs have the primary sequence characteristics of known enzymes that preferentially catalyze linalool and geraniol synthesis.

Finally, the focus of this research is unique and novel. It's very interesting research, I expect further development.

Response: Thank you for your patience and suggestions. We have improved the manuscript. The major revised portions are indicated with track changes in this revised manuscript.

Comment 1: The authors' efforts have greatly improved the presentation. In particular, this reviewer appreciates the authors' softening of the tone to match the quality of the data.

Response: Thank for your recognition to our efforts.

Comment 2: L249: "Ocimene" is usually used as a generic term. Is this trans-beta-Ocimene?

Response: It is beta-Ocimene not trans-beta-Ocimene, and we have revised that in the whole manuscript. (in red)

Comment 3: cis-allocimene → cis-allo-Ocimene

Response: We have revised that in the whole manuscript. (in red)

Comment 4: L336: "lower" → "higher"?

Response: It is ‘lower’, the linalool and geraniol represented the highest proportions of the total monoterpene contents, accounting for ~38% and 40%, respectively, in hypochile at D8, which are lower than that in other flower parts. Please see Line 123-126. (in red)

Comment 5: L339-342: Some monoterpenes, including nerol, D-limonene, (-)-trans-isopiperitenol, and citronellol, were only detected in KOVA flower tissues while ocimene, cis-allocimene, cis-citral, and citral also were not detected (Fig.2; Fig.3).

The previous manuscript did not make this point, so I could see why I did not understand a part of the previous Discussion. It often seems that the compositions of emitted and endogenous scents do not match ( https://doi.org/10.1271/bbb.70490 and https://doi.org/10.1111/ppl.12849 ). This reviewer recommend that the authors refer to these studies and discuss them in the Discussion.

Response: We have revised that and added some appropriate references. Please see Line 271-273. (in red)

Comment 6: L521: cis-allocimene → cis-allo-Ocimene

Response: We have revised that in the whole manuscript. (in red)

Comment 7: L525: the human nose → our nose

By clarifying the criteria for judging the presence or absence of scent, it will be a more accurate description.

Response: We have revised that and added some appropriate references. Please see Line 232. (in red)

Comment 8: L692: The statement of "spatial discrepancy" contradicts the views (L376-379 and L426-429) and claims (L429-431 or L534-537) of the authors. I think a more detailed explanation is required.

Response: We have revised that in this manuscript. Please see 321-323 (in red)

Comment 9: Figure 3

The origin of the "Relative content" should be explained in the figure legend.

"Ocimene" is usually used as a generic term. Is this trans-beta-Ocimene?

Please unify the notation of compounds.

For example,

beta-farnesen → beta-Farnesen

cis-allocimene → cis-allo-Ocimene

Response: We have revised that in Figure 3 legend, please see Figure 3 legend.

Comment 10: Figure 4

The origin of the "Total relative content" and "percentage" should be explained in the figure legend.

Response: We have revised that in Figure 4 legend, please see Figure 4 legend.